# Maturational Changes Alter Effects of Dietary Phytase Supplementation on the Fecal Microbiome in Fattening Pigs

**DOI:** 10.3390/microorganisms8071073

**Published:** 2020-07-18

**Authors:** Barbara U. Metzler-Zebeli, Jutamat Klinsoda, Julia C. Vötterl, Doris Verhovsek

**Affiliations:** 1Unit Nutritional Physiology, Institute of Physiology, Pathophysiology and Biophysics, Department of Biomedical Sciences, University of Veterinary Medicine, 1210 Vienna, Austria; 2Institute of Animal Nutrition and Functional Plant Compounds, Department for Farm Animals and Veterinary Public Health, University of Veterinary Medicine, 1210 Vienna, Austria; Jutamat.Klinsoda@vetmeduni.ac.at (J.K.); julia.voetterl@vetmeduni.ac.at (J.C.V.); 3Institute of Food Research and Product Development, University of Kasetsart, Bangkok 10900, Thailand; 4University Clinic for Swine, Department for Farm Animals and Veterinary Public Health, University of Veterinary Medicine, 1210 Vienna, Austria; doris.verhovsek@vetmeduni.ac.at; 5vetFarm, University of Veterinary Medicine, 1210 Vienna, Austria

**Keywords:** age, bacterial clusters, feed intake, feces, microbiome, pig, phytase

## Abstract

Age-related successions in the porcine gut microbiome may modify the microbial response to dietary changes. This may especially affect the bacterial response to essential nutrients for bacterial metabolism, such as phosphorus (P). Against this background, we used phytase supplementation (0 or 650 phytase units/kg complete feed) to alter the P availability in the hindgut and studied the dietary response of the fecal bacterial microbiome from the early to late fattening period. Fecal DNA were isolated after 0, 3, 5 and 10 weeks and the V3-V4 region of the 16S rRNA gene was sequenced. Permutational analysis of variance showed distinct bacterial communities for diet and week. Alpha-diversity and taxonomy indicated progressing maturation of the bacterial community with age. *Prevotellaceae* declined, whereas *Clostridiaceae* and *Ruminococcaceae* increased from weeks 0 to 3, 5, and 10, indicating changes in fiber-digesting capacities with age. Phytase affected all major bacterial taxa but reduced species richness (Chao1) and diversity (Shannon and Simpson). To conclude, present results greatly support the importance of available P for bacterial proliferation, including fibrolytic, lactic acid- and butyrate-producing genera, in pigs. Results also emphasize the necessity to assess bacterial responses to dietary manipulation at several time points throughout the fattening period.

## 1. Introduction

The gastrointestinal microbiota largely relies on the host animal for the supply of nutrients. Depending on the bacterial substrate preferences, the microbiota compete with the host for easily digestible nutrients (e.g., starch and protein), whereas by fermenting dietary fiber the microbes convert a non-usable dietary component into absorbable nutrients in the form of short-chain fatty acids (SCFA) [1]. Besides macronutrients, the gut microbiota rely on mineral supply originating from the diet or intestinal secretions and largely compete with the host if the provision becomes low [2]. Especially phosphorus (P) is an essential macroelement for the gut microbiota; bacteria will downregulate cell replication and metabolism, if this mineral becomes deficient [3]. For this reason, sufficient supply of the rumen microbiota with P has long been recognized as crucial in order not to compromise rumen fiber fermentation and SCFA production [3]. Evidence is accumulating that the gastrointestinal P availability affects the gut microbiome composition in monogastric livestock species as well, thereby affecting the intestinal abundance of opportunistic pathogens [4,5]. While mammals cannot utilize phytate-P, the major storage form of P in plants, gut bacteria comprise the necessary enzymatic capacities [6]. However, gut microbes only turn on this machinery when P in their surroundings becomes very low [6]. Due to ecological and economic reasons, dietary levels of inorganic P have been reduced in pig production [7] and different feed technological measures are taken to improve the availability of phytate-P for monogastric farm animals, modifying the microbial availability of P along the gastrointestinal tract [8,9]. The most common strategy is the addition of microbial phytase to diets for pigs, for which we could recently show to have a large impact on the ileal and fecal bacterial communities in weaned pigs, affecting the most abundant taxa (i.e., *Clostridiaceae* and *Lactobacillaceae*), as well as pathobionts, like *Helicobacter*, *Campylobacter* and *Fusobacterium* [9,10]. 

It is often assumed that the maturational changes in the gastrointestinal microbiome become less pronounced the more time elapses after weaning and that the microbiome composition is more affected by the diet. Nevertheless, age-related changes are still present [11] possibly influencing the response of the intestinal microbiota to dietary changes. Dietary responses of the gut microbiota in growing pigs are often only assessed at one time point, typically at the termination of the experiment. If the gut microbiota progressively changes throughout the fattening period, it can be assumed that the microbial responses to diets may change over time. Since intestinal digesta and feces are a potential reservoir of food-borne pathogens for pork contamination during the slaughtering process [12,13], age-related differences in dietary effects on the gut bacterial composition are valid to investigate in order to optimize dietary formulations and reduce pathogen transmission. We therefore hypothesized that maturational changes in the microbiota composition would alter the bacterial response to a diet supplemented with phytase and hence to changes in the intestinally available P when comparing the microbiota composition in the early, mid and late fattening period. Furthermore, reports on different ‘community types’ in pig populations [14] also led us assume that inter-individual differences in the gut microbiota composition may alter the microbial responses towards the applied phytase supplementation and intestinal P availability. Although feces may not be the suitable material in all situations, they allow multiple samplings in one animal over time, thereby reducing the inter-animal variability. In this study, we used phytase supplementation to alter the P availability in the hindgut and investigated the dietary response of the fecal bacterial microbiome from the early to late fattening period.

## 2. Materials and Methods 

### 2.1. Diets

To study the effect of phytase supplementation, age and feed intake on diet-related changes in the fecal microbiome, 2 diets were formulated that met or exceeded the current recommendation for nutrient requirements [15,16], except total P which was marginally lower than recommended for the body weight (BW) of pigs at the beginning of the experiment. Diets consisted of wheat, barley and soybean meal and were either not supplemented with phytase (control diet) or supplemented with phytase (phytase diet; Table 1). The basal diet was prepared as 1 batch with all ingredients except the premixes. The basal diet was divided, with each half being mixed with the respective premix without or with microbial 6-phytase (Garant Tiernahrung GmbH, Pöchlarn, Austria). In corresponding to common supplementation levels [6] and to account for potential losses due to pelleting, the phytase diet was formulated to contain 650 phytase units (FTU)/kg complete feed. Both diets were pelleted after mixing. Several diet samples were taken during the course of the experiment, and homogenized before using subsamples for nutrient analyses. The analyzed phytase activity (LUFA Nord-West, Oldenburg, Germany) was 257 and 596 FTU/kg complete feed in the control and phytase diets, respectively. 

### 2.2. Animals and Housing

All animal experimentation procedures were approved by the institutional ethics committee of the University of Veterinary Medicine (Vienna, Austria) and the national authority according to paragraph 26 of Law for Animal Experiments, Tierversuchsgesetz 2012 – TVG 2012 (GZ 68.205/0221-WF/V/3b/2017). The crossbred litters (*n* = 12/litter, except 1 litter with 11 pigs; Large White × Piétrain) of 6 sows were used in this experiment. Pigs were weaned with 28 days of age and penned as intact litter groups during the rearing period. At the age of 11 weeks, siblings were moved and group housed per litter in pens equipped with Feed Intake Recording Equipment (FIRE) feeders (Schauer Agrotonic, Wels, Austria). Each pen (17 m^2^; length 5.26 m × width 3.26 m) had concrete slatted floor. For environmental enrichment, each pen was equipped with one wooden block hanging from a chain, a plastic ball and a hedgehog-shaped toy. Additionally, 2 fresh sisal ropes per pen were provided at least once a week. Three pens (litters) received the control diet, whereas the other 3 pens (litters) received the phytase diet via the FIRE feeders. Water (via two nipple drinkers) and feed were freely available throughout the rearing and experimental period. During the experimental period, the individual feed intake was recorded daily, while BW was recorded bi-weekly. Fecal samples for microbiome analysis were aseptically collected from all pigs via rectal stimulation in the morning of day 2 (week 0), 16 (week 3), 23 (week 3), 36 (week 5) and 72 (week 10) of the experimental period. As the intestinal passage takes longer than 24 h, it can be assumed that the influence of the new diet on the bacterial composition on the second experimental day was negligible. The fresh feces were immediately homogenized, placed in cryo-tubes and frozen at −80 °C. Except for the first sampling (week 0), the individual feed intake for the 4 days prior to the fecal sampling of week 3, 5 and 10 was averaged to assess the impact of the feed intake on the fecal microbiome composition. The health status of all pigs were monitored daily by visual inspection and by checking the feed intake records. 

### 2.3. Chemical Analyses

Prior to proximate analysis, diets were homogenized and ground through a 1-mm screen (GRINDOMIX GM200 and Ultra-Zentrifugalmühle ZM200, Retsch, Haan, Germany). Proximate nutrient analysis including the determination of dietary Ca, P and total starch was performed according to VDLUFA [17]. Analysis of α-amylase-stable NDF exclusive of residual ash and ADF exclusive of residual ash was done using Fibre Therm FT 12 (Gerhardt GmbH & Co. KG, Königswinter, Germany) with heat-stable α-amylase [18]. 

### 2.4. DNA Extraction and 16S rRNA Gene Amplification

Only pigs (control diet, *n* = 30 pigs; phytase diet, *n* = 34) for which fecal samples from all sampling days were collected and which did not need any medical treatment throughout the experiment were included in the microbiome analysis. From these pigs, total DNA was isolated from fecal samples as described previously [19,20] using the PowerSoil DNA isolation kit (Qiagen, Hilden, Germany). The extracted DNA was quantified using quantitative PCR with primers and amplification conditions described in Klinsoda et al. [9]. Amplicon sequencing of the V3-V4 region of the 16S rRNA gene was performed on an Illumina MiSeq sequencing v2 platform using a paired-end protocol by a commercial provider (Microsynth, Balgach, Switzerland). This included the 16S rRNA PCRs, library preparation, and sequencing. The primers 341F-ill (5′-CCTACGGGNGGCWGCAG-3′) and 802R-ill (5′-GACTACHVGGGTATCTAATCC-3′) were used for the library preparation (Nextera XT DNA Sample Preparation Kit, Illumina), which generate an amplicon of approximately 460 bp. The 16S rRNA PCRs were performed using the KAPA HiFi HotStart PCR Kit (Roche, Baden, Switzerland). Subsequently, reads were demultiplexed and adapter sequences were removed using cutadapt (https://cutadapt.readthedocs.org/). Raw sequencing data can be found at the NCBI BioProject databank (PRJNA622914).

### 2.5. Bioinformatic Analysis and Statistical Analysis

Sequencing data were analyzed using the DADA2 package (version 1.12.1) in R studio (version 1.0.136; Boston, MA, USA). After inspecting the quality profiles of the forward and reverse reads separately, the first 10 nucleotides for each read were trimmed and the total length of reads were truncated to 220 nucleotides to account for the large decrease in quality score observed thereafter. Moreover, all reads containing any ambiguities were removed as were reads exceeding the probabilistic estimated error of 2 nucleotides. After de-replication of the filtered data, error rates were estimated and amplicon sequence variants (ASV) were inferred [21]. The DADA2 method infers ASVs exactly without imposing arbitrary thresholds. This allows resolving ASVs that differ by as little as one nucleotide. Thereafter, the inferred forward and reverse sequences were merged. Thereby, paired sequences that did not perfectly match were removed to control against residual errors, and a sequence table was built. After chimera removal using the removeBimeraDenovo() function, taxonomy was assigned using the Greengenes database (version 13_8) with a dissimilarity threshold of 3%. For α-diversity (Shannon, Simpson, Chao1) analysis, the samples were rarified to an equal library size using the ‘rarefy_even_depth()’ function in phyloseq. Cluster analysis to identify ‘community types’ within the fecal samples was performed using k-means clustering and the kmeans() and pam() functions in the cluster R package. First, the k number of clusters (from 2 to 10) with the highest average Silhouette width (SI) among all combinations of pairs of β-diversity measures was selected, with the score above an SI threshold of 0.25 [22]. Permutational analysis of variance (PERMANOVA) was used to statistically assess dissimilarity matrices (Bray-Curtis) derived from the microbiome data for sex, pen (litter), age, diet and ‘community-type’ using the adonis2 function in the vegan R package. Clustering of fecal samples according to sex, pen (litter), age, diet and ‘community type’ was visualized in two-dimensional nonmetric multidimensional scaling (NMDS) ordination plots obtained with the ‘metaMDS’ function in the vegan R package [23]. 

To identify the most discriminant genera that were influenced by pig’s feed intake, multigroup supervised DIABLO N-integration networking was performed using the mixOmics R package (version 6.3.2) [24] as described previously [9,19]. Sparse partial least square regression enabled the discrimination of genera across treatment groups with the lowest possible error rate, selecting 10 genera and associated them with the individual feed intake for each sampling time point. Only the strongest associations after 3, 5 and 10 weeks on the diets were projected using relevance networking and the ‘network’ function in mixOmics. 

Variables of feed intake, taxa abundances and α-diversity indices were tested for normal distribution by the Shapiro–Wilk test with the UNIVARIATE procedure in SAS (Version 9.4, SAS Inst. Inc., Cary, NC, USA). If not normally distributed, log-transformed data were used for subsequent statistical analysis in SAS. First, repeated measures were used to assess differences in feed intake, relative taxa abundances and α-diversity indices among sampling days and diets using the MIXED procedure in SAS. Second, to compare differences between dietary groups at the different sampling days, data were subjected to ANOVA using the MIXED procedure in SAS. The first random model included the fixed effects of sex and diet and their two-way interaction and pen as random effect. Pig was the experimental unit. Except for the start of the experiment, females consumed less feed than males from 3 weeks on the diet. This effect was consistent throughout the experiment and no interactive effects of sex and diet were observed. Moreover, results for the major taxa abundances and α-diversity indices were compared when using feed intake as covariate and without this covariate. As results hardly showed differences, data for females and males were analyzed together. The degrees of freedom were approximated by the Kenward-Rogers method (ddfm = kr). The means were reported as least-squares means ± standard error of the mean (SEM). Comparisons between least squares means were performed using the Tukey-Bonferroni correction. Differences were considered significant if *p* ≤ 0.05 and as trend if *p* ≤ 0.10. 

For differences in the taxonomic composition for the individual sampling days, the ‘DESeq’ function within the DESeq2 package (version 1.14.1) [25] in R was used to test for differentially abundant taxa by diet. In doing so, the ‘DESeq’ function models raw counts using a negative binomial distribution and adjusts internally for ‘size factors’ which normalize for differences in sequencing depth between sample libraries. In addition, the taxonomic datasets were pre-filtered to keep only features that have at least 10 reads total using the R command in DESeq2 “rowSums(counts(deseq_data)) ≥ 10” to remove low-count taxa and functions for the phylum through ASV level analysis. DESeq2 default settings were used to replace and filter for count outliers. Differential taxa and function abundance between diets were identified using the “Wald” test [25]. Data were listed as normalized read counts per feature. The correction of *p* values relating to the taxonomic profiles were performed using the Benjamini–Hochberg false discovery rate (FDR) [26].To account for the multiple comparisons at each taxonomic and functional level, we considered a type I error rate of ≤0.05 and a FDR-adjusted *p* value (*q* value) ≤ 0.10 as significant. Mean counts for diet groups were computed using the “sapply” function in DESeq2.

## 3. Results

### 3.1. Feed Intake

Except the dietary level of phytase, the macro- and micro-nutrient concentrations were similar between diets (Table 1). Pigs that were selected for the microbiome analysis ate on average 2.03, 2.13, and 2.57 ± 0.036 kg feed per day during the four days before feces collection in weeks 3, 5 and 10, respectively (Table 2). Except for the start of the experiment, females consumed less feed than males after 3 weeks on the diet (*p* < 0.05). Feed intake was similar among pig groups in weeks 3, 5 and 10.

### 3.2. Microbiota Composition Analysis and Age-Related Changes

Total bacterial 16S rRNA gene abundance was similar across all fecal samples, irrespective of diet, sampling time point, pen (litter) and sex. Illumina MiSeq sequencing of the V3-V4 region of the 16S rRNA gene generated a total of 6,447,814 reads for the 256 samples, with an average sequencing depth of 25,187 reads per sample and a mean read length of 418 bp. Since most ASVs were not annotated at species level, results are primarily presented at genus level. Since the sex of the pig was found to have a non-significant effect on the fecal microbiome community in our experiment (PERMANOVA; Table 3; Figure 1a), it was not considered in the further analysis. The PERMANOVA indicated an influence of pen and hence a litter effect (*p* = 0.001; Table 3). However, this effect disappeared (*p* = 0.48) when diet and week were included in the PERMANOVA (Table 3). 

Moreover, the PERMANOVA revealed an age (week) effect on the fecal microbiome (Table 3; Figure 1b), which was supported by α-diversity measures and taxonomy. Species richness declined from week 0 to week 3, 5 and 10 of the experimental period, whereas the bacterial diversity (Shannon and Simpson) fluctuated but overall declined when comparing week 0 with week 10 (Table 4). Age-related changes affected more or less the abundance of all major phyla (Appendix A). Accordingly, *Firmicutes* as the most abundant phylum increased (*p* < 0.05) from 60.2 to 64.1% of all reads from week 0 to 3 and remained at this level afterwards. *Bacteroidetes* (on average 28.2% of all reads) and *Spirochaetes* (on average 3.4% of all reads), the next most abundant phyla, declined and increased from week 0 to week 10, respectively (*p* < 0.05). Similar to phylum level, age-related alterations in taxonomy affected the majority of genera with a mean relative abundance above 0.05% (Appendix A), including the dominant taxa. Accordingly, predominant *Prevotella* declined by 0.37-fold, whereas genera belonging to *Clostridiaceae* and *Ruminococcaceae* increased by 0.70 and 0.24-fold, respectively (*p* < 0.001), from week 0 to week 10 of the experiment. Other genera showed fluctuations in their abundances over the sampling time points. For instance, *Lactobacillus* increased by 0.32-fold from week 0 to 3, returned back to the initial abundance in week 5 and increased again by 0.51-fold from week 5 to 10 (*p* < 0.05). 

The PERMANOVA showed a diet effect across sampling weeks as well as a significant diet × week interaction (*p* = 0.001; Table 3; Figure 1c). Starting with a similar bacterial diversity and composition at all taxonomic ranks between diets in week 0 (Appendix A), the phytase affected the bacterial community composition, species richness differently at the consecutive sampling time points. After 3 weeks on the diets, species richness (Chao1) of the fecal microbiome increased in pigs fed the phytase diet compared to pigs fed the control diet (*p* < 0.05; Table 4), whereas the overall diet effect (*p* = 0.014) indicated a reduced Simpson index for the bacterial communities in pigs fed the phytase diet compared to pigs fed the control diet in weeks 3 and 5. Species richness and evenness were equal between diets in week 10. 

Like for the diversity, diet effects differed at the various taxonomic ranks after 3, 5 and 10 weeks, with the greatest effects being observed at 3 weeks. Correspondingly, samples of pigs fed the phytase diet comprised more *Firmicutes* (+0.64-log_2_fold) and *Proteobacteria* (+1.01-log_2_fold) after 3 weeks on the diets compared to those of pigs fed the control diet (*p* < 0.01; Appendix A), whereas this effect disappeared after 5 weeks on the diet and turned in a decreasing effect of phytase on *Proteobacteria* abundance after 5 and 10 weeks on the diet. Only after 5 weeks on the diets, the phytase supplementation further decreased the abundance of *Spirochaetes* (*p* < 0.05). Phytase-related changes in fecal taxonomy were further demonstrated at genus level, where the phytase supplementation increased several dominant genera within the families *Clostridiaceae* and *Ruminococcaceae* including *Clostridium* and *Oscillospira* from week 3 through week 10 compared to the control diet (*p* < 0.05; Table 5). Other genera were only increased by the phytase at one but not at all three sampling time points, such as *Oscillospira* within the family *Ruminococcaceae* and *Turicibacter* after 3 and 5 weeks on the diet, respectively. Contrastingly, the phytase decreased the fecal numbers of *Prevotella* and *Megasphaera* from week 3 to week 10, whereas it reduced the abundances of *Lactobacillus*, *Roseburia*, *Streptococcus, Dialister, Treponema, Blautia* and an unclassified *S24-7 genus*, compared to the control diet only for one or two but not for all three sampling time points (*p* < 0.05).

### 3.3. Cluster Analysis

Cluster analysis was used to identify fecal samples that had per se a different microbiome structure within the current pig population. Using this method, we identified two major bacterial clusters in the fecal samples over the four sampling time points (Figure 1d). Specifically, the microbiomes of 25 samples clustered apart. These bacterial communities comprised less *Prevotella*, *Megasphaera, Streptococcus* and *Lactobacillus* but more *Faecalibacterium* and *Escherichia* compared to the other fecal samples (Cluster B; *p* < 0.05; Table 6). Although the majority of these samples were different at the start of the feeding period (week 0), it was not that whole litters clustered apart but only some animals per litter comprised a different bacterial community. The differences in the two ‘community types’ disappeared mostly after 3 weeks on the experimental diets; only in few animals (*n* = 3 to 4) this persisted after 3, 5 or 10 weeks on the experimental diets. Moreover, species richness (Chao1) and evenness (Shannon and Simpson) were higher (*p* < 0.05) in the *Faecalibacterium-Escherichia*-enriched community (cluster B) compared to cluster A (Appendix A). Despite differences in the actual abundances at the start of the experiment, the ‘community-type’ did not modulate the diet response across time points (PERMANOVA; Table 3).

### 3.4. Associations between Bacterial Genera in Feces and Pig’s Feed Intake

To characterize the influence of pig’s actual feed intake (also being indicative for pig’s total P intake) on the bacterial community, sPLS regression and relevance networking was used to identify bacterial genera that associated with pig’s feed intake after 3, 5 and 10 weeks on the diets. Only the strongest significant associations are presented (Figure 2a–c). After 3 weeks on the diets, an unclassified *RFP13* genus and *Escherichia* negatively associated with the feed intake. Two weeks later, these relationships were replaced by negative associations of *Lactobacillus*, *Streptococcus* and *p-75-a5* with the feed intake and, after 10 weeks on the diets, by four negative (unclassified *Mogibaccteriaceae*, *Paludibacter, RF17* and *Bacteroides*) and one positive (*Dialister*) associations.

## 4. Discussion

Albeit maturational changes in the porcine gut microbiome become smaller the more time passes after weaning [11,27], present results demonstrate significant fluctuations in the fecal microbiota composition and diversity from the early to late fattening period. Corresponding to our findings in young growing pigs [9], phytase affected, positively or negatively, all major bacterial genera in the present study, representing approximately 50% of all reads. This strongly supports the importance of P for bacterial metabolism [28]. By assuming a similar situation in the large intestine, this may have impacted the degradation of feed residuals and SCFA production as well as may have altered the innate immune response in the distal large intestine [29]. Our results also showed that the phytase-associated alterations in the fecal microbiome composition differed over time, indicating that the observed maturational changes modified the response of the fecal microbiome to the dietary manipulation.

Although the representativeness of the fecal community for the porcine gut microbiota is often questioned, monitoring the bacterial community over time and thus multiple samplings on the same (intact) animal can only be achieved when using feces as the gut material of choice [27]. The bacterial microbiome composition is qualitatively similar to those in the distal parts of the large intestine [11]. Hence, fecal microbiome data add valid information to our understanding of nutrient–microbiota interactions. Our data also provide useful information in that pigs are housed in a closed environment (pens), where they take up feces while playing or deliberately, which contributes to bacterial (pathogen) transmission between animals. Likewise, fecal smears on the skin of the pig are a potential source for carcass contamination during the slaughtering process [13].

Inter-individual differences in the gut microbiome can reduce the chance to distinguish dietary effects on the gut community composition in a pig population [30]. Certain pen and hence litter-associated differences were noticeable in the fecal microbiomes throughout the course of the experiment according to the PERMANOVA, which may be related, for instance, to a maternal effect or differences in the micro-environment ‘pen’ [31]. Contrarily, the two identified ‘community types’ were not litter related. More interestingly, these differences in the bacterial community composition mainly existed at the start of the experiment and disappeared thereafter, supporting the strong impact of the diet on the gut microbiome composition [19,20,32,33]. Since the fecal microbiome composition on the second experimental day (week 0) was mainly affected by the conditions prior to the experiment, the cause for these differences remains difficult to trace back as pigs were similarly housed and treated (including feeding) from birth and kept together in litter groups postweaning. Potential reasons for the observed differences in the fecal microbiome structure might be pigs’ intake of colostrum, sow milk or solid feed pre- and postweaning. As one ‘community type’ comprised more *Prevotella*, *Lactobacillus*, *Ruminococcus* and *Megasphaera*, whereas the other was enriched in *Faecalibacterium* and *Escherichia*, differences in gut microbial–host networking can be assumed. Intestinal production of butyrate due to its multiple effects upon the host as energy source and upon apoptosis, inflammation, and oxidative stress is generally considered to be beneficial to gut health [1]. Both ‘community types’ were enriched in butyrate producers; however, *Megasphaera* and *Faecalibacterium* rely on different substrates, supporting different microbial interactions. By including lactate utilizers, the genus *Megasphaera* may have profited from cross-feeding relationships with *Lactobacillus* [1], whereas *Faecalibacterium* may have utilized pectin-rich dietary residuals or host-derived *N*-acetylglucosamine as substrate [34].

Information is available about maturational changes in the porcine gut microbiome for the fattening compared to the pre- and early postweaning phase [27,35]. The bacterial diversity is commonly used as a measure for the stability of the gut microbiota, with a higher diversity representing a greater stability [36]. Here, we found a decline in bacterial species richness (Chao1) and diversity in feces from week 11 of age (week 0 of the experiment) to week 14 of age (week 3 of the experiment), after which it remained at this level until week 21 of age (week 5 of the experiment). This was in contrast to previous observations where no changes [37] or an increase in fecal diversity between 3 and 6 months of age [11,27] were reported. Since pigs did not show signs of enteric disease and had a high growth intensity in the present study, we can assume that this loss in species richness and evenness was in the physiological range, not affecting the stability of the present fecal microbiomes. At taxonomic level, especially the changes in the predominant bacterial genera within the very versatile families *Prevotellaceae*, *Clostridiaceae* and *Ruminococcaceae* from week 0 to 10 point towards alterations in metabolic capabilities to degrade complex fibrous materials and protein with increasing age. Age-related shifts in the bacterial community likely changed cross-feeding relationships among bacteria [1]. Notably, *Megasphaera* abundances coincided with *Prevotella* abundances in the present study, suggesting metabolite dependencies between these two taxa. Other taxa showed fluctuations over time, such as *Lactobacillus* and *Coprococcus,* which may have been related to competition for substrate and mutual promotion or inhibition by other bacteria. Some age-related alterations in the fecal community may have been linked to the higher feed intake with increasing BW of the pig, which was supported by relevance networking. However, more than 50% of the variation need to be explained by other factors.

Phytase improves the P availability and absorption in the upper digestive tract, leaving less P available for microbial metabolism in the large intestine [8]. Although the present fecal microbiome composition differed from our previous study in the actual abundances [9], similar taxa were affected, showing that genera within *Clostridiaceae* and *Ruminococcaceae* largely benefited from the lower P availability in the distal large intestine due to the phytase supplementation after 3 and 5 weeks on the diets. Many of these genera, except *SMB53* within *Clostridiaceae*, were not assigned at genus or species level, making it difficult to deduce metabolic features for these genera and consequences for the host. From rumen research, evidence exists that especially taxa with cellulolytic and hemicellulolytic capabilities but less amylolytic bacteria reduce growth and metabolic activity with lower concentrations of P in their environment [3]. Both families, *Clostridiaceae* and *Ruminococcaceae,* comprise amylolytic and pullulanolytic species [20,38], which may have tolerated the lower P availability in their environment better than other fibrolytic bacteria. By being a metabolically extremely versatile genus per se, *Prevotella* is typically associated with a plant-rich diet and a microbiome enriched in this taxon has an increased potential to ferment complex polysaccharides [39]. Due to the *Prevotella* dominance in the present microbiomes, their phytase-related decline may have a stronger impact on degradation of hemicellulose and cellulose and microbe-to microbe and microbial–host interactions than lower abundant taxa. If a taxa with higher P requirement became less abundant due to maturation, more P may have been available for other taxa. This may help explaining that some taxa (e.g., *Lactobacillus, Streptococcus* and *Turicibacter*) were only affected at certain sampling time points but not at all. Also, maturational changes in host physiology and digestion during the fattening period may have contributed to the age-related responses of the fecal bacteria to the phytase. The same fattening diet was fed over the whole experimental period. Unfortunately, the apparent total tract digestibility was not determined in this study. The present results for the maturational changes in the fecal microbiome emphasize the need to include this measurement in future gut microbiome studies. However, it can be generally assumed that pigs received less nutrients than required in the early and more nutrients than required in the later stages of the fattening period [16]. As a consequence, more P was absorbed in the small intestine, intensifying the P depletion in large intestinal digesta in the early compared to the later stages of the fattening period. Simultaneously, changes in other nutrients in digesta from the early fattening period to the later stages possibly altered the nutrient profile for microbial metabolism in the large intestine over time. This may explain, at least in part, the varying abundances of *Lactobacillus*, *Streptococcus*, *Dialister*, *Turicibacter, Oscillospira, Blautia* and *Christensenellaceae*, which rely on different simple and complex carbohydrates, from week 3 to 10. Besides, phytase supplementation can increase the digestibility of cations, such as calcium [40] and amino acids [41], potentially improving their digestibility. Free Ca-ions bind to cell wall components of Gram-positive bacteria, including proteins, exopolysaccharides, and lipoteichoic acid [42,43], and hence can modify adhesive interactions with protein and polysaccharide adhesion molecules at the cell surface [44]. Especially *Lactobacillus* may have a certain requirement of free Ca ions for their attachment to the intestinal mucosa [5,45]. Therefore, lower intestinal Ca availability due to increased intestinal absorption of Ca with the phytase may help explaining, similar to our previous study [9], the lower abundance of *Lactobacillus* in feces of these pigs. Calcium and P, by forming an insoluble Ca-P complex at pH values above 5 [46], as well as peptides contribute to buffering of the intestinal lumen. Since bacteria are susceptible to changes in the environmental pH, this may be another rationale having contributed to the phytase-associated alterations in the bacterial community. In a similar manner, the lower abundance of *Treponema,* a genus with proteolytic abilities [47], with phytase may link to a reduced large intestinal protein flow. Although a recent study could not find an effect of phytase on small intestinal mucin secretion [48], the increased abundance of proteolytic and mucolytic taxa, such as *Clostridium* and *Escherichia* [49], may hint at an increased glycoprotein utilization. Since genera like *Clostridium* and *Escherichia* comprise important porcine pathobionts, it seems valid to examine the effect of phytase on expression of virulence factors in functional approaches, which could not be covered by targeting the 16S rRNA gene sequencing approach.

In conclusion, the present results provide evidence that, during the fattening phase, bacterial responses to a dietary treatment change over time, emphasizing the need to investigate dietary effects in the short and long term. Whilst the bacterial phytase responses differed from the early to late fattening period, results largely support the importance of P for bacterial proliferation in the large intestine of pigs as the phytase supplementation affected the fecal abundances of all major genera in the present study. Moreover, cluster analysis identified two bacterial ‘community types’ at the start of the experiment, indicating the importance of screening for inter-individual differences in the gut microbiome when investigating bacterial dietary responses.

## Figures and Tables

**Figure 1 microorganisms-08-01073-f001:**
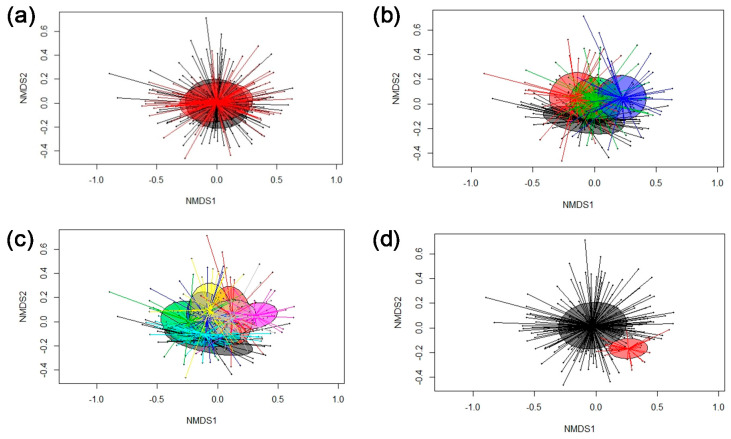
Non-metric multidimensional scaling (NMDS) plot of pairwise Bray–Curtis dissimilarities between bacterial communities in feces of pigs fed diets with (Phy diet) or without phytase (Con diet; stress level = 0.187). (**a**) sex (grey, female; male, red); (**b**) sampling time points (grey, week 0; red, week 3; green, week 5; and blue, week 10); (**c**) diet effect per time point (grey, Con diet week 0; turquoise, Phy diet week 0; green, Con diet week 3; yellow, Phy diet week 3; blue, Con diet week 5; red, Phy diet week 5; withish, Con diet week 10; pink, Phy diet week 10); and (**d**) ‘community type cluster (grey, cluster A; red, cluster B). Ellipses represent the standard deviation.

**Figure 2 microorganisms-08-01073-f002:**
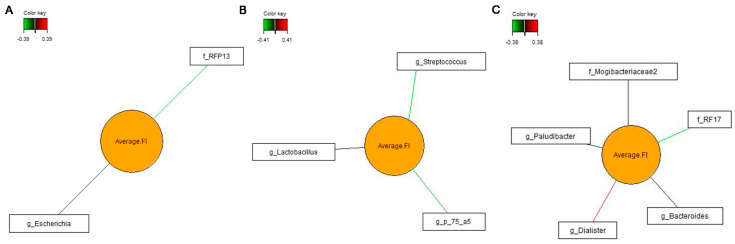
Relevance networks illustrating pairwise associations between most discriminant bacterial genera and average daily feed intake (Average_FI; |*r*| > 0.3). Covariations were calculated using sparse partial least squares regression. The relevance network is displayed graphically as nodes (bacterial genera and average feed intake) and edges (biological relationship between nodes). The edge color intensity indicates the level of the association: red = positive, and green = negative. g_, genus; f_, family. (**a**) Week 3; (**b**) week 5; and (**c**) week 10.

**Table 1 microorganisms-08-01073-t001:** Dietary ingredients and chemical composition.

Ingredients (%)	Control Diet	Phytase Diet
Barley	45.06	45.00
Wheat (11% CP)	35.51	35.46
Soybean meal HP (47% CP)	8.11	8.10
Soybean meal (42% CP)	7.01	7.00
Calcium carbonate	1.28	1.28
Rapeseed oil	1.00	1.00
Monocalcium phosphate	0.46	0.46
Salt	0.46	0.46
Lysine-HCL 98	0.41	0.41
Premix ^1,2^	0.39	0.39
l-Threonine	0.13	0.13
Magnesiumoxide	0.10	0.10
DL-Methionine	0.08	0.08
Phytase (phytase units/kg) ^3^	-	650
Analyzed chemical composition (dry matter basis)		
Dry matter	89.86	89.80
Crude ash	5.05	5.03
Crude protein	18.40	18.86
aNDF_OM_	13.82	14.07
ADF_OM_	5.75	5.73
Starch	69.20	69.10
Sugar	5.40	5.60
Calcium	7.48	7.50
Phosphorus	5.68	5.59
Calcium:phosphorus	1.32	1.34

^1^ The vitamin–mineral premix without phytase provided per kilogram of experimental diet (Garant GmbH, Pöchlarn, Austria): vitamin A 6510 IE, vitamin D_3_ 2003 IE, vitamin E 104.973 mg, vitamin K_3_ 3.005 mg, vitamin B_1_ 1.502 mg, vitamin B_2_ 4.006 mg, vitamin B_3_ 20.031 mg, vitamin B_6_ 2.003 mg, vitamin B_12_ 0.020 mg, pantothenic acid 10.016 mg, folic acid 0.501 mg, biotin 0.050 mg, choline chloride 132.172 mg, Fe 160.460 mg, Cu 21.567 mg, Zn 122.060 mg, Mn 67.358 mg, Mo 0.860 mg, J 1.721 mg, Co 0.102 mg, Se 0.568 mg. ^2^ The vitamin–mineral premix with phytase provided per kilogram of experimental diet (Garant GmbH, Pöchlarn, Austria): vitamin A 6502 IE, vitamin D_3_ 2001 IE, vitamin E 104.837 mg, vitamin K_3_ 3.001 mg, vitamin B_1_ 1.500 mg, vitamin B_2_ 4.001 mg, vitamin B_3_ 20.005 mg, vitamin B_6_ 2.001 mg, vitamin B_12_ 0.020 mg, pantothenic acid 10.003 mg, folic acid 0.500 mg, biotin 0.050 mg, choline chloride 132.000 mg, Fe 160.251 mg, Cu 21.539 mg, Zn 121.901 mg, Mn 67.270 mg, Mo 0.859 mg, J 1.719 mg, Co 0.102 mg, Se 0.567 mg. ^3^ 650000 U 6-phytase (3 VM Phytase XP 897420).

**Table 2 microorganisms-08-01073-t002:** Average feed intake of pigs fed diets with or without phytase during the four days prior to sampling ^1^.

Diet	Control		Phytase			*p*-value		
Sex	f	m	f	m	SEM	Sex	Diet	Sex × Diet
Week 3	1.91	2.25	1.89	2.18	0.054	<0.001	0.398	0.667
Week 5	2.03	2.26	2.08	2.22	0.054	0.001	0.938	0.412
Week 10	2.35	2.78	2.56	2.69	0.095	0.005	0.533	0.131

^1^ f, females; m, males. Per time point: Control diet: *n* = 19 females and 11 males per time point; phytase diet: *n* = 18 females and 16 males per time point.

**Table 3 microorganisms-08-01073-t003:** Permutational multivariate analysis of variance (PERMANOVA) results for fecal communities of pigs fed diets with or without phytase ^1^.

Covariables	df	SS	R^2^	F	*p*-Value
Cluster	1	0.83	0.05	17.41	0.001
Diet	1	0.93	0.05	19.59	0.001
Week	1	2.31	0.13	48.69	0.001
Sex	1	0.07	0.004	1.43	0.178
Pen	4	1.54	0.09	8.08	0.001
Cluster × diet	1	0.09	0.005	1.81	0.079
Cluster × week	1	0.26	0.01	5.37	0.002
Diet × week	1	0.10	0.006	2.14	0.049
Cluster × sex	1	0.02	0.001	0.48	0.830
Diet × sex	1	0.05	0.003	1.13	0.272
Week × sex	1	0.06	0.003	1.23	0.250
Cluster × pen	3	0.21	0.01	1.47	0.115
Week × pen	4	0.47	0.03	2.47	0.001
Sex × pen	4	0.31	0.02	1.65	0.046
Cluster × diet × week	1	0.04	0.002	0.81	0.531
Cluster × diet × sex	1	0.03	0.002	0.58	0.742
Diet × week × sex	1	0.04	0.002	0.83	0.494
Cluster × week × pen	2	0.07	0.004	0.76	0.653
Cluster × sex × pen	2	0.06	0.003	0.66	0.783
Week × sex × pen	4	0.17	0.009	0.88	0.575
Residual	219	10.44	0.58		
Total	255	18.12	1.00		

^1^ The analysis based on pairwise distance of a multivariate data set and values were obtained using type III sums of squares with 999 permutations of residuals, considering significant difference at *p* ≤ 0.05. df, degrees of freedom; SS, sum of squares.

**Table 4 microorganisms-08-01073-t004:** Alpha diversity metrices in feces of pigs fed diets with or without phytase supplementation ^1^.

Week	0	0		3	3		5	5		10	10		*p*-value		
Diet	Con	Phy	SEM	Con	Phy	SEM	Con	Phy	SEM	Con	Phy	SEM	Week	Diet	Week × Diet
Chao1	663	618	23.2	413b	503a	23.2	484	530	23.2	470	455	23.2	<0.001	0.376	0.005
Shannon	5.320	5.267	0.049	4.960	5.053	0.049	5.092	5.102	0.049	4.939	4.891	0.049	<0.001	0.912	0.305
Simpson	0.984	0.983	0.002	0.982 ^a^	0.978 ^b^	0.002	0.983 ^a^	0.977 ^b^	0.002	0.976	0.972	0.002	<0.001	0.014	0.446

^1^ Con, control diet; Phy, phytase diet. Per time point: Control diet: *n* = 30 per time point; phytase diet: *n* = 34 per time point. ^a,b^ Different lowercase letters per time point within a row indicate significant difference (*p* < 0.05).

**Table 5 microorganisms-08-01073-t005:** Differentially abundant bacterial genera in feces of pigs fed diets with or without phytase supplementation over time.

	Week 3					Week 5					Week 10				
Genus	Mean ^1^	log_2_ Fold Change ^2^	SE	*p*-Value	*q*-Value ^3^	Mean ^1^	log_2_ Fold Change ^2^	SE	*p*-Value	*q*-Value ^3^	Mean ^1^	log_2_ Fold Change^2^	SE	*p*-Value	*q*-Value ^3^
*Prevotella*	4129	−0.86	0.191	<0.001	<0.001	3307	−0.60	0.205	0.003	0.018	2041	−0.66	0.228	0.004	0.017
*Clostridiaceae*	1358	0.75	0.206	<0.001	0.003	1739	0.67	0.206	0.001	0.009	1751	0.14	0.160	0.397	0.569
*Ruminococcaceae*	1350	0.51	0.128	<0.001	0.001	1559	0.31	0.118	0.009	0.041	1217	0.26	0.081	0.001	0.009
*Lactobacillus*	1223	−1.02	0.420	0.015	0.044	928	−0.68	0.424	0.111	0.242	1047	−0.61	0.342	0.075	0.175
*SMB53*	902	0.64	0.182	<0.001	0.004	957	0.48	0.171	0.005	0.023	763	0.01	0.147	0.945	0.961
*Megasphaera*	722	−1.39	0.240	<0.001	<0.001	412	−1.83	0.384	<0.001	<0.001	147	−2.37	0.657	<0.001	0.004
*Oscillospira*	705	0.69	0.177	<0.001	0.001	864	0.18	0.138	0.199	0.373	646	0.21	0.088	0.016	0.051
*Ruminococcus*	576	0.25	0.186	0.175	0.324	619	−0.29	0.157	0.068	0.165	361	−0.01	0.139	0.951	0.961
*Treponema*	568	−0.60	0.230	0.009	0.033	726	−0.54	0.217	0.013	0.051	597	−0.06	0.235	0.809	0.863
*S24-7*	532	−0.33	0.198	0.097	0.206	699	0.20	0.187	0.283	0.480	694	−0.43	0.137	0.002	0.010
*Lachnospiraceae*	470	−0.14	0.199	0.489	0.643	568	−0.06	0.193	0.742	0.854	512	−0.22	0.175	0.214	0.373
*Roseburia*	431	−0.96	0.316	0.003	0.011	258	−1.08	0.311	<0.001	0.007	150	−0.76	0.382	0.046	0.118
*Bacteroidales*	415	0.08	0.157	0.631	0.755	580	0.42	0.166	0.012	0.049	517	0.40	0.159	0.011	0.040
*Veillonellaceae*	385	−1.36	0.332	<0.001	0.001	216	−1.27	0.412	0.002	0.012	57	−1.67	0.569	0.003	0.016
*Streptococcus*	375	−1.19	0.362	0.001	0.006	608	−0.24	0.365	0.507	0.687	487	−0.08	0.315	0.806	0.863
*Dialister*	347	−0.98	0.372	0.008	0.033	219	−0.83	0.489	0.088	0.203	60	−1.64	0.783	0.036	0.099
*Coprococcus*	341	0.19	0.115	0.093	0.201	396	−0.12	0.118	0.320	0.520	279	−0.13	0.089	0.137	0.280
*Clostridiales*	326	0.86	0.165	<0.001	< 0.001	395	0.36	0.167	0.032	0.099	291	0.45	0.141	0.001	0.009
*Blautia*	325	−0.20	0.159	0.210	0.372	261	−0.72	0.180	<0.001	0.001	156	−1.07	0.321	0.001	0.008
*Phascolarctobacterium*	278	0.36	0.206	0.077	0.174	321	0.39	0.212	0.064	0.159	208	−0.12	0.219	0.596	0.744
*Succinivibrio*	262	0.24	0.434	0.579	0.726	150	−0.74	0.394	0.060	0.152	88	−1.52	0.450	0.001	0.007
*[Prevotella]*	259	−0.80	0.239	0.001	0.006	235	−0.27	0.218	0.208	0.384	146	−0.51	0.302	0.094	0.208
*Faecalibacterium*	243	0.13	0.273	0.628	0.755	169	−0.59	0.250	0.019	0.066	80	−1.17	0.354	0.001	0.008
*[Mogibacteriaceae]*	133	0.26	0.215	0.221	0.372	146	−0.19	0.195	0.326	0.524	113	−0.27	0.140	0.050	0.124
*Anaerovibrio*	130	−0.81	0.380	0.034	0.092	85	−0.25	0.407	0.537	0.708	46	−1.42	0.552	0.010	0.039
*RFP12*	127	−0.79	0.304	0.009	0.033	197	−0.14	0.255	0.573	0.733	196	−0.13	0.293	0.646	0.764
*Dorea*	127	−0.18	0.207	0.385	0.565	117	−0.25	0.184	0.180	0.349	71	−0.35	0.141	0.012	0.043
*Bulleidia*	123	−0.13	0.188	0.494	0.643	104	0.23	0.172	0.185	0.353	53	−0.33	0.240	0.170	0.333
*Christensenellaceae*	119	0.13	0.360	0.712	0.821	213	0.78	0.324	0.016	0.056	272	0.93	0.286	0.001	0.009
*Butyricicoccus*	115	0.31	0.179	0.083	0.184	115	−0.04	0.195	0.832	0.906	68	0.09	0.220	0.684	0.791
*[Ruminococcus]*	113	−0.12	0.222	0.595	0.731	88	−0.40	0.207	0.056	0.148	48	−0.64	0.261	0.014	0.046
*CF231*	112	−0.29	0.226	0.197	0.359	145	−0.16	0.224	0.464	0.663	127	−0.86	0.203	<0.001	0.001
*Sarcina*	111	0.61	0.380	0.109	0.220	105	0.24	0.312	0.439	0.651	72	−0.20	0.394	0.605	0.744
*Peptostreptococcacae*	104	0.66	0.219	0.003	0.011	65	0.46	0.184	0.013	0.051	46	−0.21	0.224	0.342	0.529
*[Eubacterium]*	97	−0.46	0.223	0.040	0.099	66	−0.17	0.227	0.443	0.651	27	−0.75	0.217	0.001	0.007
*Turicibacter*	97	0.47	0.286	0.104	0.215	131	0.78	0.234	0.001	0.008	117	0.22	0.186	0.242	0.400
*Clostridium*	49	1.69	0.568	0.003	0.013	113	1.20	0.494	0.015	0.055	154	0.04	0.442	0.927	0.961
*Parabacteroides*	41	0.85	0.438	0.053	0.125	70	0.86	0.330	0.009	0.041	45	0.58	0.256	0.023	0.064
*[Paraprevotella]*	38	−0.18	0.298	0.552	0.708	57	−0.25	0.221	0.264	0.467	40	0.00	0.205	0.986	0.986
*Escherichia*	35	2.74	0.759	0.000	0.003	40	0.76	0.456	0.095	0.215	35	0.33	0.436	0.446	0.629

Normalized reads (hit counts). Only the most abundant genera pathways (>0.01% of the mean hit counts on at least one of the three sampling days) between the two dietary groups are presented. Con, control diet; Phy, phytase diet. Per time point: Con diet: *n* = 30; Phy diet: *n* = 34. ^1^ Mean normalized reads across diets per time point. ^2^ Standard error of the log_2_ fold change. ^3^ False discovery rate (Benjamini–Hochberg) corrected *p*-value.

**Table 6 microorganisms-08-01073-t006:** Differences in selected genera (hit counts) between the two fecal ‘community type’ clusters in pigs.

Genus	Mean ^1^	log_2_ Fold Change ^2^	SE	*p*-Value	*q*-Value ^3^
*Prevotella*	4411	−1.13	0.223	<0.001	<0.001
*Clostridiaceae*	1747	−0.06	0.169	0.712	0.794
*Ruminococcaceae*	1592	0	0.069	0.948	0.965
*Lactobacillus*	1297	−1.28	0.350	<0.001	0.001
*SMB53*	959	−0.40	0.142	0.005	0.013
*Oscillospira*	791	−0.21	0.100	0.032	0.068
*S24-7*	759	0.11	0.145	0.452	0.590
*Megasphaera*	713	−2.90	0.444	<0.001	<0.001
*Treponema*	680	0.04	0.190	0.845	0.883
*Bacteroidales*	606	0.62	0.115	<0.001	<0.001
*Ruminococcus*	569	−0.44	0.135	0.001	0.004
*Streptococcus*	550	−1.54	0.324	<0.001	<0.001
*Faecalibacterium*	548	0.66	0.163	<0.001	<0.001
*Roseburia*	471	−1.73	0.347	<0.001	<0.001
*Lachnospiraceae*	423	−0.25	0.112	0.028	0.060
*RFP12*	421	−2.06	0.465	<0.001	<0.001
*CF231*	385	0.30	0.121	0.012	0.029
*Blautia*	331	−1.04	0.224	<0.001	<0.001
*Phascolarctobacterium*	328	−0.24	0.182	0.190	0.294
*[Prevotella]*	306	0.29	0.230	0.200	0.302
*Dialister*	302	−4.98	0.488	<0.001	<0.001
*Coprococcus*	243	−1.35	0.295	<0.001	<0.001
*Escherichia*	219	1.10	0.253	<0.001	<0.001
*Succinivibrio*	213	−1.35	0.383	<0.001	0.002
*Christensenellaceae*	201	0.30	0.316	0.344	0.469
*Anaerovibrio*	160	−1.08	0.404	0.008	0.020
*Bulleidia*	154	0.65	0.188	0.001	0.002
*[Mogibacteriaceae]*	152	0.43	0.133	0.001	0.005
*Clostridium*	142	1.24	0.412	0.003	0.008
*Dorea*	133	−0.29	0.162	0.071	0.132
*Butyricicoccus*	128	−0.28	0.176	0.107	0.175
*Turicibacter*	126	0.22	0.211	0.289	0.414
*Veillonellaceae*	126	0.19	0.460	0.673	0.777
*Peptostreptococcacae*	116	−0.99	0.196	<0.001	<0.001
*[Ruminococcus]*	107	−1.49	0.230	<0.001	<0.001
*Mitsuokella*	92	−3.40	0.599	<0.001	<0.001
*[Eubacterium]*	92	−1.31	0.258	<0.001	<0.001
*Sarcina*	89	−1.54	0.464	0.001	0.003
*Pirellulaceae*	88	−0.30	0.184	0.105	0.173
*Oribacterium*	68	−2.18	0.250	<0.001	0.000
*Coriobacteriaceae*	67	−0.33	0.242	0.176	0.277
*RFN20*	66	−0.12	0.221	0.572	0.679
*Campylobacter*	64	−26.11	1.632	<0.001	<0.001
*Parabacteroides*	62	0.08	0.303	0.785	0.851
*Clostridiales*	58	−1.88	0.572	0.001	0.004
*Catenibacterium*	56	−3.95	0.555	<0.001	<0.001
*RF16*	55	1.84	0.487	<0.001	0.001
*BS11*	54	2.02	1.031	0.051	0.099
*Acidaminococcus*	54	−5.52	1.137	<0.001	<0.001
*WPS-2*	52	0.88	0.444	0.047	0.094
*Lachnospira*	52	−0.97	0.338	0.004	0.012
*Paludibacter*	52	0.95	0.310	0.002	0.007
*[Paraprevotellaceae]*	51	−0.16	0.193	0.416	0.549
*Methanobrevibacter*	51	−0.42	0.452	0.352	0.475
*p-75-a5*	48	0.72	0.204	<0.001	0.002
*Shuttleworthia*	48	−3.77	0.781	<0.001	<0.001
*Collinsella*	44	−1.26	0.327	<0.001	0.001

Normalized reads (hit counts). Only the most abundant genera (>0.2% of the mean hit counts) are presented (*n* = 8 per diet). Log_2_ fold changes were calculated by comparing cluster B with cluster A. Cluster A: *n* = 231 samples; Cluster B: *n* = 25 samples. ^1^ Mean normalized reads across ‘community type’ clusters. ^2^ Standard error of the log_2_ fold change. ^3^ False discovery rate (Benjamini–Hochberg) corrected *p*-value.

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
