# Peer review of "Maturational Changes Alter Effects of Dietary Phytase Supplementation on the Fecal Microbiome in Fattening Pigs"

_microorganisms, 2020, doi:10.3390/microorganisms8071073_

Round 1
Reviewer 1 Report
The manuscript is presenting interesting data on the effects of dietary phytase supplementation on the fecal microbiome in fattening pigs. The paper is well written, clearly demonstrating the aim of the study.
There are no important changes to be done before publishing, at least not in the important aspects of the study and the presentation of the results and discussions.
- I would complete the first sentence in the abstract (as is somehow bold... and the host).
- also, I would organize the abstract in Introduction/material and methods/results/conclusions
- see line 217-219 - the text should not be left in the paper
Author Response
Dear reviewer,
thank you very much for your comments and positive assessment of our manuscript.
Best regards,
Barbara Metzler-Zebeli
- I would complete the first sentence in the abstract (as is somehow bold... and the host).
AUTHORS: We modified the first sentence of the abstract.
- also, I would organize the abstract in Introduction/material and methods/results/conclusions
AUTHORS: Thank you for this suggestion. The abstract was actually organized accordingly, without explicitly using the terms due to the limit of 200 words.
- see line 217-219 - the text should not be left in the paper
AUTHORS: Thank you. Indeed, this sentence should not be left in the paper.
Reviewer 2 Report
First I'd like to congratulate authors for carrying out the experiment and very good description, especially good is discuss. Results obtained from microbiom investigation are usually very difficult to interpret due to the abundance of data, especially when we use the second generation sequencing technologies.
And now some comments:
a) Fig 1 b - what is for?, it is unnecessary I recommend to delete it
b) Fig 2 - I don't like it (by the way it is not too clear). I don't understand why you didn't measure FCR (feed conversion ratio) what is more related to microbiom (and important for us) especially when we have no to big differences in feed intake. Feed intake could be more connected with taste of feed.
Once again I'd like to underline that I like your discuss part but in future I recommend to add body weight and FCR or some health parameters (APP or others) to use them for better interpretation.
Additionally:
line 30: it is written profileration - should be proliferation
line 217-219 - remove it
line 473 - remove the first sentence from Author Contributions
Author Response
Dear reviewer, thank you very much for your general assessment of our work and helpful comments.
Best regards,
Barbara Metzler-Zebeli
Specific comments:
- Fig 1 b – what is for?, it is unnecessary I recommend to delete it
AUTHORS: Thank you for indicating that this fig. 1b couuld be removed. We modified Figure 1 accordingly.
- Fig 2 - I don't like it (by the way it is not too clear). I don't understand why you didn't measure FCR (feed conversion ratio) what is more related to microbiom (and important for us) especially when we have no to big differences in feed intake. Feed intake could be more connected with taste of feed.
AUTHORS: Thank you for your criticism. However, there seem to be a misunderstanding. This study was not aimed to investigate relationships between feed-efficiency and the fecal microbiome in order to identify marker bacteria (we have done this previously in pigs and chickens using residual feed intake as metric for feed efficiency (Siegerstetter et al., 2018 or McCormack et al., 2017). With respect to this figure, we tried to understand whether certain bacteria may be linked to the feed intake of the pigs and thus to pig’s P intake. This seemed reasonable as the feed intake of the pigs increased over time.
Once again I'd like to underline that I like your discuss part but in future I recommend to add body weight and FCR or some health parameters (APP or others) to use them for better interpretation.
AUTHORS: Thank you for this suggestions.
Additionally:
line 30: it is written profileration - should be proliferation
AUTHORS: corrected.
line 217-219 - remove it
AUTHORS: Thank you. Removed.
line 473 - remove the first sentence from Author Contributions
AUTHORS: Thank you. Apologies that we missed to delete this.